# Occupational Hearing Loss Associated with the Combined Exposure of Solvents and Noise: A Systematic Review and Meta-Analysis

**Jia Ren [1,2], Hongwei Xie [1], Yong Hu [1], Yu Hong [2,\*], Hua Zou [1,\*] and Xiangjing Gao [1,\*]**

[1] Department of Occupational Health and Radiation Protection, Zhejiang Provincial Center for Disease Control and Prevention, Hangzhou 310051, China; renjia202310@163.com (J.R.); hwxie@cdc.zj.cn (H.X.); huyong@cdc.zj.cn (Y.H.)

[2] Department of Public Health, Hangzhou Normal University, Hangzhou 310030, China

\* Correspondence: xjgao@cdc.zj.cn (X.G.); hzou@cdc.zj.cn (H.Z.); hongyu_xj@126.com (Y.H.); Tel.: +86-0571-87115127 (H.Z.); +86-0571-87115213 (X.G.)

**Abstract:** To analyze the predominant frequencies of hearing threshold shift and the prevalence of hearing loss related to the co-exposure to noise and solvents. A systematic review and meta-analysis were performed by retrieving published articles from Web of Science, PubMed, Scopus, Embase, and ProQuest until July 2023. Data were extracted in line with the Cochrane Collaboration Handbook, and the Newcastle-Ottawa Scale and Agency for Healthcare Research and Quality were used to assess the studies' quality. The meta-analysis was used to estimate the odds ratios (ORs) with 95% confidence interval (CI). $I^2$ and Q statistics were used to prove the heterogeneity. A total of 22 selected studies (9948 workers), six cohort studies and 16 cross-sectional studies were included. The results revealed that 43.7%, 41.3%, and 53.6% of the participants presented with hearing loss due to noise exposure, solvent exposure, and combined exposure to noise and solvent, respectively. The workers exposed to both noise and solvents had a higher risk of hearing loss than those exposed to noise (overall weighted odds ratio [OR]: 1.76) or solvents (overall-weighted OR: 2.02) alone. The poorer hearing threshold in the combined noise and solvents exposure group was mainly at high frequencies (3, 4, 6, and 8 kHz), with a peak of 29.47 dB HL at 6 kHz. The noise-exposed group's peak hearing threshold was 28.87 dB HL at 4 kHz. The peak hearing threshold of the solvent-exposed group was 28.65 dB HL at 6 kHz. The workers exposed to noise and solvent simultaneously had a higher prevalence of hearing loss than those exposed to solvents. Co-exposure to noise and solvents increases the odds of hearing loss. The dominant hearing threshold changes occurred at 3, 4, 6, and 8 kHz, and the peak value appeared at 6 kHz in workers co-exposed to noise and solvents.

**Keywords:** hearing loss; combined exposure; ototoxic solvents; noise effects; occupational exposure



## 1. Introduction

Hearing loss is a prevalent sensory disability worldwide. On the basis of the World Health Organization's estimation, by 2050 nearly 2.5 billion (one in four) people will have some degree of hearing loss, and at least 700 million will need rehabilitation [1]. Occupational hearing loss is a prevalent occupational disease globally. In developed countries, more than 10% of workers suffer from hearing loss [2]. Occupational noise-induced hearing loss (NIHL) is the most widespread occupational disease in the United States (US) [3], and was estimated to account for more than 60 percent of all occupational diseases in Norway [4]. Similarly, occupational noise-induced deafness is the second-most common diagnosed occupational disease in China [5]. In addition, occupational hearing loss has been listed as a priority research area in the 21st century by the US National Institute for Occupational Safety and Health [6].

Previously, noise exposure was universally recognized as the only risk factor for hearing loss in occupational settings. With the development of occupational health research, other ototoxic factors associated with hearing loss in industrial environments, such as organic solvents, have been revealed [7]. Organic solvents are liquid compounds, with low molecular weight and high volatility, which are widely used in diverse industries, including the industry of shoes, furniture, dyes, adhesives, plastic, rubber, electronics, and printing [8]. Hearing loss induced by solvent exposure has been demonstrated in animals since the late 1980s [9]. Lataye and Campo reported that exposure to organic solvents is related to hearing loss [10]. Sensorineural hearing loss with cochlear damage has been demonstrated in animals exposed to solvents such as toluene, styrene, xylene, n-hexane, trichloroethylene, ethyl benzene, and white spirits [11]. Meanwhile, epidemiological studies on the effects of solvents have been growing, which have provided the evidence that trichloroethylene, carbon disulfide, toluene, styrene, and a mixture of solvents could enhance hearing loss [12–16].

The combined exposure to noise and solvents has become prevalent. Some surveys have found that the combined exposure of organic solvents and noise is common in the furniture manufacturing industry and printing industry [11,17,18]. According to the statistics, there are more than 590,000 furniture manufacturing enterprises, and 98,000 printing enterprises in 2020, and the concentration of organic solvents in many of these enterprise exceeded the occupational exposure limits [19–22]. In the last few decades, the influence of combined exposure has been extensively investigated. Early in 1984, the enhanced hearing loss in individuals exposed to noise and solvents compared to those exposed to noise alone was first found by Barregard and Axelsson [16]. A later study in a larger population confirmed that the exposure to noise combined with organic solvents could lead to excessive hearing loss [10]. Morata et al. [18] and Morioka et al. [23] revealed that exposure to solvents at concentrations under the limits recommended by international agencies could be harmful to hearing. The association between the exposure to organic solvents and NIHL was confirmed by Vyskocil et al. [24]. Moreover, some studies have reported that the prevalence of hearing loss in the populations exposed to a noise level below the occupational exposure limit may increase after co-exposure to organic solvents [25–27]. Furthermore, three literature reviews were conducted on the impact of combined solvents and noise exposure on hearing loss [28–30]. Hodgkinson et al. [28] conducted a review on the relationships between the risk of hearing loss and occupational exposure to solvents alone or combined solvents and noise exposure. The results showed that workers with organic solvents exposure had an increased risk of hearing loss. Moreover, as reported by Hormozi et al. [30], the odds of hearing loss in workers exposed to a combination of noise and solvents were significantly higher than in those exposed to solvents alone. However, the ORs for workers with combined solvents exposure and noise exposure alone have not been compared. Another systematic review by Nakhooda et al. [29] compared the ORs of workers among three exposure groups, but excluded studies that used participants with hearing loss and no other auditory dysfunctions. In addition, the systematic review from Golmohammadi et al. investigated the combined effects of co-exposure to occupational noise and other factors (including solvents). This review reported that the level of evidence for the combined effects of noise and solvents was high [31].

Despite substantial studies and previous reviews demonstrating that simultaneous exposure to noise and solvents can enhance hearing loss, the dominant frequencies of hearing threshold shift and the prevalence of hearing loss associated with co-exposure to solvents and noise have not been clarified. Moreover, recently published epidemiological studies with larger sample sizes enable a more detailed review. Therefore, the aims of this review are as follows: (i) to analyze the prevalence of hearing loss associated with the exposure to solvents and noise, and highlight the implication of combined exposure on hearing loss for decision-makers, and (ii) to analyze the predominant frequencies of hearing threshold shift related to solvents alone or in combination with noise.

## 2. Materials and Methods

### 2.1. Literature Retrieval

The following English literature databases were used: Web of Science, PubMed, Scopus, Embase, and ProQuest. The keywords used for searching were "hearing loss", "audiology", "hearing disorder", "hearing impairment", "hearing threshold shift", "noise-induced hearing loss", "NIHL", "solvents", "ototoxicity chemical", and "noise". In addition, the list of references for other relevant papers has been reviewed. The search was completed in July 2023.

### 2.2. Inclusion and Exclusion Criteria

The inclusion criteria were: (1) studies were designed as cohort, case-control, or cross-sectional studies and published in English journals; (2) studies were conducted on human beings alone; (3) studies included participants with co-exposure to solvents and noise; and (4) studies provided OR value or relative risk with the corresponding 95% confidence intervals (CIs).

The exclusion criteria were: (1) in vitro studies on the mechanisms of hearing loss; (2) animal experiments conducted on co-exposure to noise and solvents; (3) studies with unclear results or unclear description of participants; (4) studies in a language other than English; and (5) books, reviews, and conference papers.

### 2.3. Data Extraction

The relevant literature was screened and extracted using EndNote software (version X9.1; Thomson ResearchSoft Corp., Stanford, CT, USA). Information regarding the first author, year of publication, area, industry type, sample size, noise level, types of solvents, exposure duration, hearing threshold, the prevalence of hearing loss, and general information about the target population was extracted from each study for systematic review and meta-analysis. Two investigators (R.J. and G.X.J.) independently extracted information from the eligible studies according to Cochrane Collaboration Handbook [32]. In this review, the outcome is hearing loss and is defined as the average hearing threshold of binaural or monaural above 25 dB at speech frequency (0.5, 1, and 2 kHz), or at high frequency (3, 4, 6, and 8 kHz), or hearing threshold above 25 dB HL at 0.5,1, 2, 4, 6, and/or 8 kHz.

### 2.4. Quality Assessment

The methodological quality of the included cross-sectional studies and cohort studies were assessed using the 11-item checklist recommended by the Agency for Healthcare Research and Quality (AHRQ) and Newcastle-Ottawa Scale (NOS), respectively. For the AHRQ method, an item would be scored '0' if it was answered 'NO' or 'UNCLEAR'; if it was answered 'YES', then the item scored '1'. Articles scoring 0–3 points, 4–7 points, and 8–11 points were classified as low, moderate, and high-quality studies, respectively. For the NOS, the judgment was based on three areas: selection of the participants, comparability of groups, and exposure/outcome ascertainment. Scores ranging from 0 to 9 indicate an improvement in the quality of the studies.

### 2.5. Statistical Analysis

ORs with 95% CIs were used to describe the pooled estimates for outcomes. The primary comparison was the risk of hearing loss in the group with noise and solvent co-exposure versus the group with noise-only or solvent-only exposure. $I^2$ and Q statistics were used to prove the heterogeneity. A fixed-effects model was used when the *p*-value of the Q test was <0.10 and the $I^2$ value was <50%. Otherwise, the random effects model was used. Furthermore, the hearing loss prevalence among the groups was compared using *t*-test, and the hearing thresholds of left/right ear at each frequency (0.5, 1, 2, 3, 4, 6, and 8 kHz), speech frequency, and high-frequency among the study groups was compared by the analysis of variance. A two-tailed *p*-value < 0.05 was considered statistically significant except for the heterogeneity test, where a *p*-value < 0.10 (two-tailed) was used. Meta-regression analysis

was used to explore the relationship between independent variables (including age, level of solvents, level of noise, and exposure duration) and hearing loss, and to screen out the influencing factors leading to the heterogeneity of effect size. The publication bias was assessed with the Egger test. All data were analyzed using the Review Manager (Version 5.4, The Cochrane Collaboration, London, UK) and SPSS software (version 16.0; SPSS Corp., Chicago, IL, USA).

## 3. Results

### 3.1. Literature Retrieval

According to the databases and search terms, 1089 studies were initially identified through the database search. After removing the duplicates, 429 articles were included. After checking the titles or abstracts according to the exclusion criteria, 346 articles were excluded. By reviewing the full text and assessing the quality, 60 items were excluded from the remaining studies. Finally, 23 articles were included (Figure 1), including 17 cross-sectional studies (73.9%) and 6 retrospective cohort studies (26.1%) on exposure to noise and solvents.

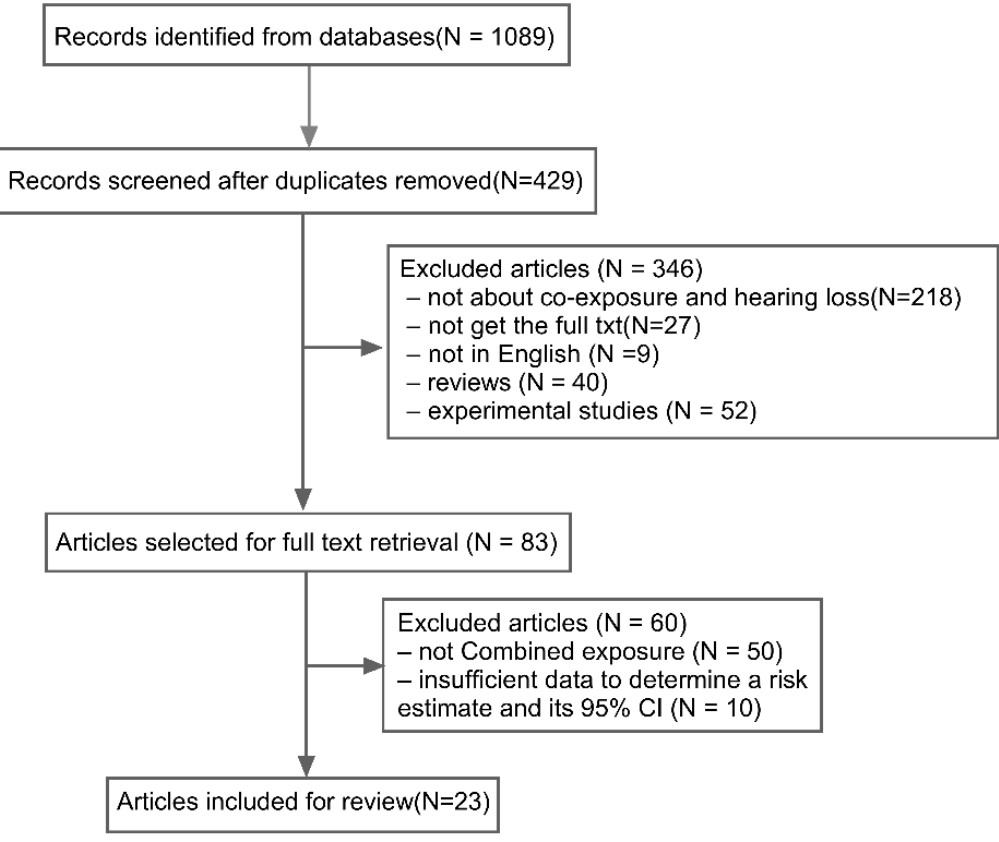

**Figure 1.** Flow chart of study selection for the meta-analysis.

### 3.2. Quality Assessment and Meta-Regression

The quality of the cross-sectional studies and cohort studies was assessed according to the AHRQ method (Supplementary Table S1) and the NOS (Supplementary Table S2), respectively. The AHRQ evaluation indicated that 16 and 1 cross-sectional studies appeared to have moderate and good quality, respectively. The NOS assessment showed that four and two cohort studies appeared to have moderate and good quality, respectively. Hence, 17 cross-sectional studies and 6 cohort studies were included in the systematic review. The result of the Egger test showed that the t = 1.57 and $p = 0.134$, which suggested that there was no significant publication bias among the included articles.

The result of the meta-regression showed that the organic solvent exposure concentration, level of noise, exposure duration, and age had no statistical significance.

### 3.3. Information about the Included Studies

According to Table 1, six cohort studies with 1406 participants were included. The years of follow-up ranged from 1 to 7 years. The mean age and exposure time of the participants were $40.8 \pm 7.7$ and $12.5 \pm 7.0$ years, respectively. Seventeen cross-sectional studies with 9712 participants were included to review (Table 2). The mean age and exposure time of the participants in cross-sectional studies were $37.4 \pm 7.8$ and $11.8 \pm 6.7$ years, respectively. The included participants were classified into four groups: noise, co-exposure, solvents, and reference (not exposed to noise or solvents) groups. The maximum level of noise for the noise, co-exposure, solvents, and reference groups was 107.5 dB(A), 105.5 dB(A), 80.3 dB(A), and 78.5 dB(A), respectively. Sixteen studies were conducted on exposure to a mixture of solvents, including benzene, toluene, xylene, styrene, and acetone. Seven studies were conducted on the exposure to one kind of solvent: malathion, $CS_2$, toluene, epoxy adhesives, ethylbenzene, and xylene isomers.

**Table 1.** Information of the cohort studies included in the meta-analysis.

| | Study | Country | Industry | Participants | | $L_{ex}$[dB(A)] | Organic Solvents | Exposure Time, y (M ± SD) | Years of Follow-Up |
|---|---|---|---|---|---|---|---|---|---|
| | | | | N | Age, y (M ± SD) | | | | |
| 1. | Guida et al. (2010) [33] | N.A. | Agriculture | 80 | A : 38.3 ± 3.5 <br> B : 39.1 ± 4.4 | 98.5 | B: Malathion (N.A) | A: 12.1 ± 5.8 <br> B: 12.6 ± 5.4 | 1 |
| 2. | Barba et al. (2005) [34] | Brazil | Petrochemical plants | 172 | 44.3 | 79.1 | B: Benzene (0.43 ppm), Toluene (0.05 ppm), Xylene (0.05 ppm), Butadiene (12.25 ppm); <br> C: Benzene (0.10 ppm), Toluene (0.05 ppm), Xylene (0.05); | N.A. | 5 |
| 3. | Lobato et al. (2014) [35] | Brazil | Chemical products | 99 | A : 39.3 ± 10.3 <br> B : 38.7 ± 8.9 | A: 89 <br> B: 93 | B: Toluene, Xylene, Turpentine, Oils, Greases, Lead Chromates and Molybdates (N.A) | A: 9.0 ± 6.5 <br> B: 9.2 ± 5.8 | 1 |
| 4. | Sliwinska-Kowalska et al. (2001) [36] | Poland | Chemical products | 517 | B: 38.4 ± 9.1 <br> C: 39.3 ± 9.5 <br> D: 38.5 ± 10.6 | B: N.A. <br> C: N.A. <br> D: 83 | B: Xylene (28.3 mg/m$^3$), Ethylacetate (7.7 mg/m$^3$), Whitespirit (7.0 mg/m$^3$), Toluene (5.8 mg/m$^3$), Butyl acetate (1.8 mg/m$^3$), Ethyl benzene (7.9 mg/m$^3$); <br> C: Xylene (28.7 mg/m$^3$), Ethyl acetate (11.5 mg/m$^3$), White spirit (11.7 mg/m$^3$), Toluene (8.4 mg/m$^3$), Butyl acetate (8.3 mg/m$^3$), Ethyl benzene (7.7 mg/m$^3$) | B: N.A. <br> C: 12.2 ± 8.5 <br> D: 12.8 ± 8.2 | 7 |
| 5. | Chang et al. (2003) [37] | China (Taiwan) | Textile | 346 | A: 42.2 ± 5.8 <br> B: 48.3 ± 8.7 <br> D: 42.0 ± 6.2 | A: 86.5 <br> B: 85.5 <br> D: 78.5 | B: CS$_2$ (N.A) | A: 12.1 ± 5.7 <br> B: 20.8 ± 10.5 <br> D: 11.3 ± 6.4 | 5 |
| 6. | Schaper et al. (2008) [38] | Germany | Printing | 192 | N.A. | A: 79 <br> B: 84 | B: Toluene (N.A) | N.A. | 5 |
| 7. | Total | / | / | 1406 | 40.8 ± 7.7 | 85.6 | / | 12.5 ± 7.0 | / |

M: mean; SD: standard deviation; A: noise-exposed group; B: noise and solvents combined exposure group; C: solvent-exposed group; D: reference group, without noise or solvent exposure; N.A.: not available.

**Table 2.** Information of the cross-sectional studies included in the meta-analysis.

| | Study | Country | Industry | Participants | | $L_{ex}$[dB(A)] | Organic Solvents | Exposure Time, y (M ± SD) |
|---|---|---|---|---|---|---|---|---|
| | | | | N | Age, y (M ± SD) | | | |
| 1. | Blair et al. (2021) [39] | USA | Air Force base | 870 | N.A. | N.A. | Benzene, Ethylbenzene, Toluene, P-Xylene (N.A) | 8.7 ± 3.1 |
| 2. | Hughes et al. (2013) [40] | USA | Air Force Reserve | 503 | 66% > 35 | A: 90 <br> B: N.A. <br> C: N.A. | B,C: Toluene, Xylene, Benzene, Styrene (N.A) | N.A. |
| 3. | Yang H-Y, et al. (2016) [41] | China (Taiwan) | Stone-processing | 314 | 51.3 ± 8.5 | A: 91.3 <br> B: 87.1 <br> C: 80.1 | B,C: Epoxy adhesives (N.A) | A: 16.5 ± 10.6 <br> B: 19.0 ± 10.6 <br> C: 20.2 ± 10.7 |
| 4. | Botelho et al. (2009) [42] | Brazil | Steel | 155 | A: 30.5 ± 6.8 <br> B: 31.8 ± 7.5 | A: 90 <br> B: 90 | B: Acetone, Styrene, Resins, and Cobalt (N.A) | A: 7.6 ± 3.5 <br> B: 6.1 ± 3.3 |

Table 2. *Cont*.

| | Study | Country | Industry | Participants N | Participants Age, y (M ± SD) | $L_{ex}$[dB(A)] | Organic Solvents | Exposure Time, y (M ± SD) |
|---|---|---|---|---|---|---|---|---|
| 5. | Metwally et al. (2012) [43] | Egypt | Painting | 222 | A: 44.1 ± 9.0<br>B: 43.5 ± 10.9<br>D: 41.5 ± 8.7 | A: 87.1<br>B: 84.7<br>D: 76.0 | B: Toluene (165.67 mg/m$^3$), Xylene (256.67 mg/m$^3$), Ethylacetate (1160.5 mg/m$^3$), Butanol (238 mg/m$^3$), Isopropranolol (458 mg/m$^3$), Acetone (1121 mg/m$^3$), Ethanol (1412.3 mg/m$^3$) | A: 20.5 ± 11.9<br>B: 18.4 ± 10.3<br>D: N.A. |
| 6. | Sliwinska-Kowalska et al. (2004) [44] | Poland | Dockyard | 906 | A: 42.2 ± 9.3<br>B: 37.4 ± 9.2<br>D: 39.8 ± 9.3 | A: 90.3<br>B: 94.2<br>D: 74.1 | B: Xylene (245.2 mg/m$^3$), Toluene (28.9 mg/m$^3$) | N.A. |
| 7. | Morata et al. (1993) [18] | Brazil | Printing | 190 | A: 36.1 ± 8.2<br>B: 32.5 ± 7.9<br>C: 31.7 ± 7.2<br>D: 34.7 ± 9.8 | A: 92.5<br>B: 93<br>C: N.A.<br>D: N.A. | B,C: Toluene (31.45 ppm), Xylene (19.64 ppm), Benzene (0.73 ppm), Methyl ethyl ketone (20.4 ppm), Ethanol (11.6 ppm), Methyl isobutyl ketone (10.25 ppm) | A: 11.6 ± 7.8<br>B: 8.1 ± 6.2<br>C: 5.6 ± 3.7<br>D: 13.1 ± 7.6 |
| 8. | Ikuharu et al. (2000) [23] | Japan | Plastic manufacturing | 54 | A: 33.2 ± 11.1<br>B: 33.8 ± 9.0<br>D: 43.6 ± 15.1 | A: 84<br>B: 72.5<br>D: 60 | B: Styrene (22.4 ppm), Methanol (23.7 ppm), Methyl acetate (24.6 ppm) | N.A. |
| 9. | Kim et al. (2005) [45] | Korea | Aviation | 328 | A: 31.2 ± 6.1<br>B: 39.6 ± 4.7<br>C: 38.6 ± 6.0<br>D: 31.3 ± 6.3 | A: 93<br>B: N.A.<br>C: N.A. | B,C: Methyl ethyl ketone (62.68 ppm), Toluene (0.81 ppm), Xylene (0.57 ppm), Methyl isobutyl ketone (0.22 ppm) | N.A. |
| 10. | Ikuharu et al. (2014) [46] | Thailand | Manufacturing | 199 | A: 35.8 ± 7.8<br>B: 36.3 ± 6.0<br>D: 37.6 ± 8.1 | A: 83.7<br>B: 84.0<br>D: 59.5 | B: Styrene (1.1 ppm), Acetone (1.1 ppm) | A: 8.5 ± 4.4<br>B: 9.2 ± 3.3<br>D: 8.0 ± 4.1 |
| 11. | Chang, Chen, Lien, and Sung (2006) [47] | China (Taiwan) | Chemical products | 176 | A: 41.5 ± 3.1<br>B: 40.0 ± 9.7<br>D: 40.9 ± 3.4 | A: 86.8<br>B: 82.9<br>D: 70.3 | Toluene (N.A) | A: 11.5 ± 5.7<br>B: 12.3 ± 8.8<br>D: 9.5 ± 5.3 |
| 12. | Jacobsen et al. (1993) [27] | Copenhagen | N.A. | 3284 | 62.9 ± 5.1 | N.A. | N.A | N.A. |
| 13. | Yuewei Liu et al. (2015) [48] | China | Waste Landfill | 247 | 38.0 ± 11.0 | B: 66.2<br>C: 64.3 | B: Volatile organic (3.4 mg/m$^3$)<br>C: Volatile organic (0.55 mg/m$^{3)}$ | 11.0 ± 8.9 |
| 14. | Sliwinska-Kowalska et al. (2003) [11] | Poland | Plastic manufacturing | 513 | A: 41.0 ± 8.4<br>B: 36.5 ± 8.2<br>C: 33.8 ± 9.1<br>D: 39.6 ± 9.7 | A: 89.2<br>B: 88.6<br>C: 80.3<br>D: 73.2 | B: Styrene (34.4 mg/m$^3$), toluene (28.0 mg/m$^3$)<br>C: Styrene (59.9 mg/m$^3$), toluene (3.4 mg/m$^3$) | N.A. |
| 15. | Mohammadi et al. (2010) [49] | Iran | Automobile | 441 | A: 33.4 ± 6.9<br>B: 33.5 ± 6.2<br>C: 31.9 ± 5.5 | A: 84.0<br>B: 84.3<br>C: N.A. | B: Benzene (0.003 mg/m$^3$), Toluene (19 mg/m$^3$), Xylene (137 mg/m$^3$), Acetone (101 mg/m$^3$)<br>C: Benzene (2.01 mg/m$^3$), Toluene (31 mg/m$^3$), Xylene (388 mg/m$^3$) | A: 8.5 ± 4.9<br>B: 8.1 ± 3.7<br>C: 7.4 ± 3.4 |
| 16. | Rizk and Sharaf. (2010) [50] | Egypt | Fermentation | 140 | A: 28.0 ± 7.1<br>B: 30.2 ± 4.9<br>D: 31.3 ± 5.6 | A: 107.5<br>B: 105.5<br>D: N.A. | B: Toluene, Xylene, Butyl acetate, Ethyl alcohol (N.A) | N.A. |
| 17. | Zhang et al. (2013) [51] | China | Petrochemical | 1170 | A: 38.2 ± 9.8<br>B: 39.5 ± 8.7<br>D: 37.0 ± 5.1 | A: 84.3<br>B: 83.1<br>D: 67.3 | Ethylbenzene (N.A) | A: 16.9 ± 10.1<br>B: 17.3 ± 9.2 |
| 18. | Total | / | / | 9712 | 37.4 ± 7.8 | 82.9 | / | 11.8 ± 6.7 |

M: mean; SD: standard deviation; A: noise-exposed group; B: noise and solvents combined exposure group; C: solvent-exposed group; D: reference group, without noise or solvent exposure; N.A.: not available.

### 3.4. Prevalence of Hearing Loss in Different Exposure Groups

A total of 11,118 workers were included, with an average exposure duration of 11.9 years. Their average noise exposure level was 81.6 dB(A), and the average age was 40.8 years (Table 3). The noise exposure group included 3115 participants, and 44.9% (n = 1398) presented with hearing loss. The solvent exposure group comprised 1006 participants, and 41.3% (n = 415) presented with hearing loss. The co-exposure group included 3537 participants, and the prevalence of hearing loss was 57.8% (n = 2044). The hearing loss prevalence in the noise- (t = 2.15, $p < 0.05$) or solvents-exposed group (t = 1.67, $p < 0.05$) was significantly higher than that in the control group (24.7%). The hearing loss prevalence in the noise and solvents combined exposure group was significantly higher than that in the noise- (t = 2.6, $p < 0.05$) and solvents-exposed groups (t = 2.2, $p < 0.05$). Figure 2 presents the hearing loss prevalence in the noise-exposed, solvents-exposed, and co-exposure groups. In 16 of the 23 articles that included the co-exposure and noise-exposed groups, a higher hearing loss prevalence was reported in the co-exposure group than in the noise-exposed group. In 10 articles that included the co-exposure and solvent-exposed groups, a higher hearing loss prevalence was reported in the co-exposure than in the solvent-exposed group.

**Table 3.** Summary of the prevalence of hearing loss in the different exposure groups.

| | Group | Population | | | | Noise Level (mean) $L_{Aeq}$ [dB(A)] | Hearing Loss (%) |
|---|---|---|---|---|---|---|---|
| | | N | Men (%) | Mean Age (Years) | Mean Exposure Duration (Years) | | |
| 1. | Noise-exposed | 3115 | 2306 (74.0) | 39.3 | 11.9 | 88.7 | 44.9 |
| 2. | Noise and solvents co-exposure | 3537 | 2430 (68.7) | 39.5 | 12.4 | 86.2 | 57.8 |
| 3. | Solvent-exposed | 1006 | 860 (85.5) | 40.7 | 10.9 | 76.0 | 41.3 |
| 4. | Reference | 3460 | 3157 (91.2) | 43.8 | 12.5 | 75.6 | 24.7 |
| 5. | Total/Mean | 11,118 | 8753 (78.7) | 40.8 | 11.9 | 81.6 | 42.2 |

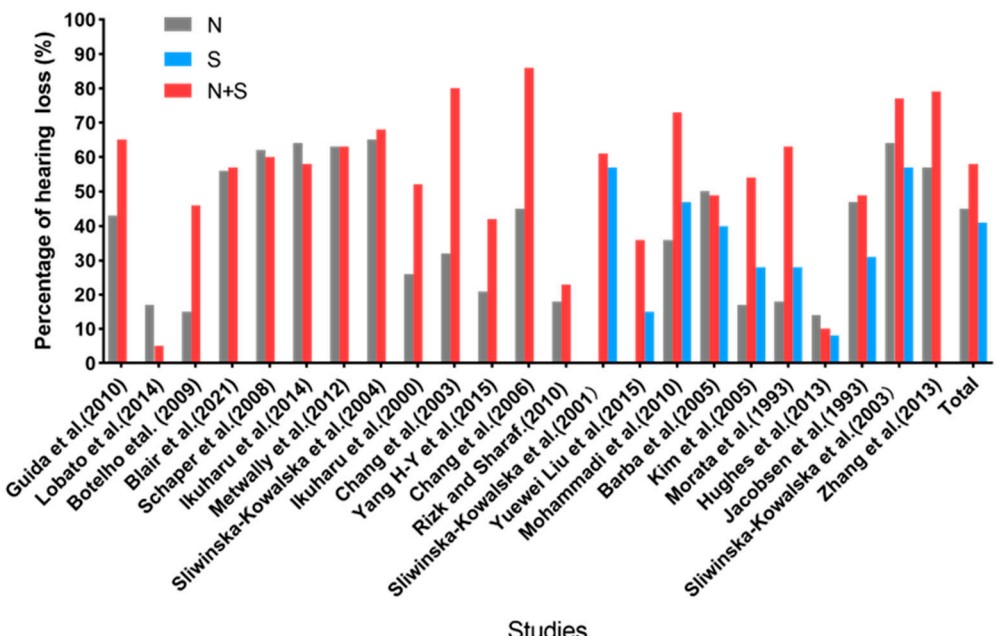

**Figure 2.** Prevalence of hearing loss among the four exposure groups in each article [11,18,23,27,33–51].

### 3.5. Comparison of the Risk of Hearing Loss between the Combined Exposure and Noise Exposure Groups

Twenty one studies with noise-exposed populations and noise and solvent co-exposed populations were investigated. Figure 3 illustrates the pooled OR values of hearing loss

between the noise-exposed and co-exposed groups in each study. The random effects model of the meta-analysis showed that the weighted OR value of noise- and solvent-combined-exposure as a risk factor for hearing loss was 1.33 (95% CI: 1.15–1.53). Among the 21 studies, the 95% CI of OR in 17 studies were >1.

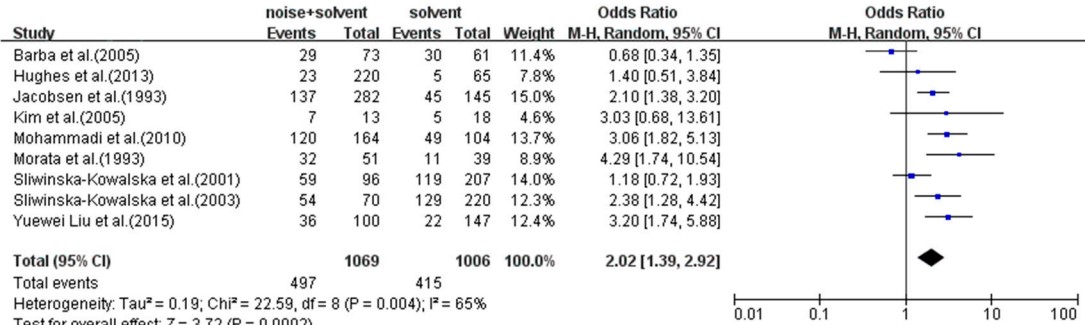

**Figure 3.** The meta-analysis forest plot of the overall weighted OR value of hearing loss in the noise and solvent combined exposure group with the noise-exposed group as the control [11,18,23,27,33–35,37–47,49–51].

### 3.6. Comparison of the Risk of Hearing Loss between the Combined Exposure and Solvents Exposure Groups

Nine studies provided information on the risk of hearing loss in the combined and solvent exposure groups. Of the nine included studies, two revealed no association, whereas others suggested a positive association between combined exposure and hearing loss. Figure 4 presents the pooled OR value of hearing loss between the solvent-exposed and co-exposure groups in each study using a random effects model. The overall weighted OR of hearing loss for the noise and solvent combined exposure as a risk factor for hearing loss was 2.02 (95% CI: 1.39–2.92).

| Study | noise+solvent Events | Total | solvent Events | Total | Weight | Odds Ratio M-H, Random, 95% CI |
|---|---|---|---|---|---|---|
| Barba et al.(2005) | 29 | 73 | 30 | 61 | 11.4% | 0.68 [0.34, 1.35] |
| Hughes et al.(2013) | 23 | 220 | 5 | 65 | 7.8% | 1.40 [0.51, 3.84] |
| Jacobsen et al.(1993) | 137 | 282 | 45 | 145 | 15.0% | 2.10 [1.38, 3.20] |
| Kim et al.(2005) | 7 | 13 | 5 | 18 | 4.6% | 3.03 [0.68, 13.61] |
| Mohammadi et al.(2010) | 120 | 164 | 49 | 104 | 13.7% | 3.06 [1.82, 5.13] |
| Morata et al.(1993) | 32 | 51 | 11 | 39 | 8.9% | 4.29 [1.74, 10.54] |
| Sliwinska-Kowalska et al.(2001) | 59 | 96 | 119 | 207 | 14.0% | 1.18 [0.72, 1.93] |
| Sliwinska-Kowalska et al.(2003) | 54 | 70 | 129 | 220 | 12.3% | 2.38 [1.28, 4.42] |
| Yuewei Liu et al.(2015) | 36 | 100 | 22 | 147 | 12.4% | 3.20 [1.74, 5.88] |
| **Total (95% CI)** | | 1069 | | 1006 | 100.0% | 2.02 [1.39, 2.92] |
| Total events | 497 | | 415 | | | |

Heterogeneity: Tau² = 0.19; Chi² = 22.59, df = 8 (P = 0.004); I² = 65%
Test for overall effect: Z = 3.72 (P = 0.0002)

**Figure 4.** The meta-analysis forest plot of the overall weighted OR value of hearing loss in the noise and solvent combined exposure group with the solvents-exposed group as the control [11,18,27,34,36,40,45,48,49].

### 3.7. The Predominant Frequencies of Hearing Threshold Shift among the Different Exposure Groups

The average hearing thresholds of the eight studies [31–34,39,42,45,47] are presented in Table 4. The analyzed threshold frequencies were 0.5, 1, 2, 3, 4, 6, and 8 kHz. The threshold

levels revealed similar trends in both ears. The average threshold did not significantly vary among the three exposure groups at each frequency. The auditory thresholds in the three exposure groups were below 15 dB HL at 0.5, 1, and 2 kHz (within the normal levels). The hearing threshold levels at 3, 4, 6, and 8 kHz (higher than 20 dB HL) were significantly higher than those at 0.5, 1, and 2 kHz. The hearing threshold levels were lower than 14 dB HL in speech frequency and higher than 24 dB HL at high frequencies in the three exposure groups. The co-exposure group demonstrated the highest hearing threshold at speech frequency and high frequency without a significant difference compared to the other two exposure groups. For both ears, the highest hearing threshold was at 4 kHz (>28 dB HL) in the noise-exposed group and at 6 kHz (>28 dB HL) in the other two exposure groups.

**Table 4.** Hearing threshold in the different exposure groups.

| Ear | | Frequency (kHz) | Noise Group | Noise and Solvents Group | Solvents Group | *p* |
|---|---|---|---|---|---|---|
| | (1) | 0.5 | 11.41 | 11.15 | 10.17 | 0.96 |
| | (2) | 1 | 13.72 | 14.39 | 14.00 | 0.96 |
| | (3) | 2 | 13.94 | 15.16 | 14.83 | 0.93 |
| | (4) | 3 | 20.98 | 21.91 | 21.88 | 0.96 |
| 1. Right | (5) | 4 | 28.86 | 27.96 | 24.71 | 0.76 |
| | (6) | 6 | 26.79 | 29.47 | 28.65 | 0.79 |
| | (7) | 8 | 24.23 | 25.69 | 24.00 | 0.91 |
| | (8) | Speech frequency (0.5, 1, and 2) | 13.02 | 13.57 | 13.00 | 0.96 |
| | (9) | High-frequency (3, 4, 6, and 8) | 25.22 | 26.26 | 24.81 | 0.85 |
| | (1) | 0.5 | 9.69 | 10.45 | 9.00 | 0.97 |
| | (2) | 1 | 12.92 | 14.65 | 13.14 | 0.85 |
| | (3) | 2 | 13.09 | 14.22 | 12.89 | 0.84 |
| | (4) | 3 | 20.42 | 21.12 | 20.88 | 0.87 |
| 2. Left | (5) | 4 | 28.87 | 26.47 | 24.65 | 0.77 |
| | (6) | 6 | 26.14 | 28.33 | 28.42 | 0.82 |
| | (7) | 8 | 21.33 | 24.32 | 24.59 | 0.58 |
| | (8) | Speech frequency (0.5, 1, and 2) | 11.90 | 13.11 | 11.68 | 0.61 |
| | (9) | High-frequency (3, 4, 6, and 8) | 24.19 | 25.06 | 24.64 | 0.91 |

## 4. Discussion

In this review, the literature on occupational hearing loss due to exposure to organic solvents and noise were assessed. The included 23 studies (with 11,118 participants) were conducted in several countries, including Brazil, Poland, China, Taiwan, Egypt, India, Iran, and the US. The participants with noise and solvent co-exposure were mainly distributed in typical manufacturing industries (automobile, ship, machinery manufacturing, and steel), which have been reported to be the primary workplaces where noise and solvents coexist [52,53]. The solvents involved in this review were mainly benzene series compounds

(toluene, ethylbenzene, styrene, and xylene), which are extensively used in industries and have been verified to be ototoxic. For example, Pryor et al. reported slightly impaired hearing loss at 8 kHz, which was markedly impaired at 12 kHz and above when male Fischer rats were exposed to toluene 14 h/day, 7 days/week, for 5 weeks [54]. The ototoxic effects of styrene and xylenes on mid-frequency hearing loss in rats were demonstrated by Pryor et al. [55]. Campo et al. revealed that ethylbenzene induces permanent hearing loss in rats [56].

This review showed that the average prevalence of hearing loss in the solvents-exposed group was 41.3%, which is significantly higher than that in the reference group. The detrimental effects of solvents on hearing have been reported in animal and human studies [54–56]. Our findings confirmed the positive effects of exposure to organic solvents on hearing loss. Another implication from this review is that the noise and solvent co-exposure group had a significantly higher prevalence of hearing loss than the noise-exposed or solvents-exposed groups. This was supported by one systematic review and meta-analysis on occupational noise-induced hearing loss in China, which has reported a higher prevalence of hearing loss (54.2%) in the co-exposure group than the noise-exposed group (30.3%) [57]. Further, the forest plot of the meta-analysis results confirmed that combined exposure to solvent and noise increased the OR estimates of hearing loss compared to the groups exposed to noise or solvent alone. The risk estimates of hearing loss in the combined exposure group were over 1.33 times higher than those in the noise exposure group and 2.02 times higher than in the solvent exposure group. Similarly, this was confirmed in the systematic review and meta-analysis on occupational noise-induced hearing loss in China, which has reported that the risk estimates of hearing loss in the combined exposure group were 2.36 times higher than that in the noise exposure group [57]. This phenomenon that the simultaneous exposure to solvents and noise increased the risk of hearing loss has been demonstrated in animal experiments. For instance, experiments with rats have revealed that the combined exposure to styrene/toluene and low noise levels induce synergistic adverse effects on hearing [10,52]. An investigation conducted by Lataye et al. reported that simultaneous exposure to noise and ethylbenzene increased the risk of hearing loss [53]. However, in human studies, the synergistic effect between solvents and noise on hearing loss has not been consistently validated owing to the differences in solvents, concentration, and exposure time [58,59]. For example, Barba et al. followed up workers who were simultaneously exposed to noise and styrene within the exposure limit for 5 years and found no significant difference in the standard threshold shift between the group with co-exposure and the group exposed only to noise [34]. Another study, which included participants who were exposed to noise and solvents exceeding the exposure limit, demonstrated that the percentage of hearing loss had a significant difference between the co-exposure and noise-exposed groups [43]. The result of our analysis confirmed that the combined exposure to noise and solvents could aggravate the risk of hearing loss.

The threshold of NIHL is most prevalent at frequencies of 3, 4, and 6 kHz, referred to as the "noise notch" [60], which is helpful for audiologists in the early diagnosis of NIHL. The solvent-induced hearing loss has been reported to be associated primarily with increased risk at higher frequencies (3, 4, 6, and 8 kHz) [36,61–63]. However, since different solvents probably have a different mechanistic interaction in inducing hearing damage, their combined exposure to noise has varying effects on hearing loss. A study of workers in fiberglass and metal product manufacturing plants exposed to noise and styrene revealed that pure-tone thresholds at 2, 3, 4, and 6 kHz were significantly lower than workers exposed to noise only [15]. Sliwinska-Kowalska et al. [36] discovered that the synergistic effect of exposure to mixed organic solvents and noise affects hearing at middle and high frequencies (3, 4, 6, and 8 kHz) in human studies. However, Chang et al. [37] discovered that hearing loss in workers exposed to carbon disulfide and noise in a viscose rayon production plant mainly occurred at speech frequencies (0.5, 1, and 2 kHz). In this review, the phenomenon that the predominantly changed thresholds occurred at 3, 4, 6, and 8 kHz in the three exposure groups was confirmed. This suggests that 3, 4, 6, and

8 kHz frequencies could be primarily affected by noise and solvent combined exposure. Remarkably, the highest threshold occurred at 4 kHz in the noise-exposed group, while in the other two exposure groups, the highest threshold occurred at 6 kHz. The primarily changed frequencies could be attributed to the pathogenetic mechanism of the solvent on hearing. Ototoxic solvents can diffuse through the outer sulcus to impair the organ of Corti and the outer hair cells (OHCs), thereby resulting in cochlear toxicity. The cochlear toxicity has been reportedly associated with the alteration of the ionic $K^+$ concentration surrounding the OHCs and the lipid peroxidation of the membrane of the OHCs in animal experiments [64–66]. Furthermore, animal studies suggested that the solvents could act on nicotinic receptors and thereby reduce the protective function of the inner-ear and middle-ear, which can block the penetration of acoustic energy [67–69]. These harmful effects of the solvents make the membrane of the OHCs more vulnerable and can disturb the protective reflexes. Thus, the combined exposure to noise and solvents is equivalent to exposure to higher noise. Therefore, the primarily changed frequencies are high frequencies, which are similar with those associated with noise.

The prevalence of hearing loss in the different exposure groups in the included studies was analyzed and demonstrated great variation. This is possibly owing to the heterogeneity of the included articles, such as different ages, levels of noise and solvents, the pathway of exposure, time of exposure, and personal protective equipment. Additionally, the difference in the definition of hearing loss is another contributing factor. The definition of hearing loss included the following aspects: thresholds above 25 dB HL at frequencies of 3, 4, and/or 6 kHz [33,35,39,40,42]; thresholds above 25 dB HL at 0.5, 1, 2, 4 and/or 8 kHz [34,37,41,44]; the average hearing threshold at 0.5, 1, and 2 kHz above 25 dB; or the average hearing threshold at 3, 4, 6, and 8 kHz above 25 dB HL [36,38,43]. However, the effects of the different definitions of hearing loss on the comparison results could be weakened for the reason that the reference groups use the same definition of hearing loss in each study. In addition, the results of the ORs in the different exposure groups confirmed the prevalence of hearing loss, which suggested that the comparison of hearing loss prevalence is credible. Certainly, the definite prevalence of hearing loss in each exposure group should be discussed without further validation being challenging.

This study had several limitations. First, the inclusion of only six cohort studies makes determining the causal relationships between co-exposure to noise and solvents and increased risk of hearing loss challenging. Second, the definition of hearing loss varied in different articles, making it impossible to standardize the prevalence of hearing loss. Third, few studies have provided explicit threshold values and the effect of gender on the threshold was not considered, which results in less accuracy in the threshold level results. Fourth, in the included studies, the exposure to organic solvents was not quantitatively assessed, which makes comparing these studies challenging. In addition, the duration of the exposure to noise and solvents in each work shift, protective measurements, and personal protective equipment, which are influential factors for the risk of hearing loss, were not available nor were they discussed in this study.

## 5. Conclusions

The results of this review indicated the contribution of noise and solvent co-exposure to hearing loss. The workers who had combined exposure to noise and solvents suffered from more significant hearing loss than those exposed to noise or solvents alone. Therefore, workers simultaneously exposed to noise and solvents should be included in hearing protection programs. Primary prevention is the most cost-effective measure, and includes methods such as reducing the level of solvents and noise, shortening the duration of exposure, and using hearing protection devices. Furthermore, it is necessary to implement important health surveillance, especially audiometry in shorter intervals, for workers exposed to organic solvents and noise. In addition, there was an indication that the hearing threshold changes at 3, 4, 6, and 8 kHz frequencies are associated with the combined exposure. This indicated that special attention should be paid to 3, 4, 6,

and 8 kHz frequencies in audiometry for workers exposed to both organic solvents and noise. However, more representative studies conducted on the characteristics of hearing thresholds associated with the combined exposure of solvents and noise are necessary to carry out. It is necessary to carry out many cohort studies to elucidate the dose–response relationship between hearing loss and co-exposure to noise and solvents.

**Supplementary Materials:** The following supporting information can be downloaded at: https://www.mdpi.com/article/10.3390/safety9040071/s1, Table S1: Quality assessment of the cross-sectional studies; Table S2: Quality assessment of the cohort studies.

**Author Contributions:** J.R. prepared the original draft and conducted the investigation. J.R., H.X. and Y.H. (Yong Hu) performed the investigation. H.X. performed the data curation. Y.H. (Yu Hong) contributed to the methodology. Y.H. (Yu Hong) conducted the formal analysis. X.G. conducted the investigation, reviewed and edited the article, and was responsible for funding acquisition. H.Z. performed the investigation and reviewed the article. All authors have read and agreed to the published version of the manuscript.

**Funding:** This study was supported by the Medical Health Technology Project by the Health Commission of Zhejiang (No. 2021KY120); the Natural Science Foundation of Zhejiang Province (No. Y22H260595) and A project Supported by Scientific Research Fund of Zhejiang Provincial Education Department (No. Y202250515).

**Institutional Review Board Statement:** Not applicable.

**Informed Consent Statement:** Not applicable.

**Data Availability Statement:** The data presented in this study are available on request from the corresponding author.

**Conflicts of Interest:** The authors declare that the research was conducted in the absence of any commercial or financial relationships that could be construed as a potential conflict of interest.

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
