# Peer review of "Occupational Hearing Loss Associated with the Combined Exposure of Solvents and Noise: A Systematic Review and Meta-Analysis"

_safety, 2023_

Round 1

Reviewer 1 Report

Authors conducted a meta-analysis of studies published in English language from 1993 on hearing disorders related to noise and solvents. 22 studies from initial 207 studies were subjected to analysis.

The result of the meta-analysis showed that the weighted OR value of noise- and solvent combined-exposure as a risk factor for hearing loss was 1.76 (95% CI:1.26–2.46).

In their discussion the authors dis not include the meta-analysis of Zhou J, Shi Z, Zhou L, et al. Occupational noise-induced hearing loss in China: a systematic review and meta-analysis. BMJ Open 2020;10:e039576. (doi:10.1136/bmjopen-2020-039576) which showed that the overall weighted OR for noise exposure as a risk factor for hearing loss was 5.63 (95% CI (CI), 4.03–7.88). Particularly this study found much more studies (88 from 594).

Also the work of Staudt et al. Int Arch Occup Environ Health. 2019 Apr; 92(3): 403–413. doi: 10.1007/s00420-019-01419-2 finding no statistically significant interaction  between these solvents and occupational noise on high-frequency hearing loss is not mentioned.

Missing is the systematic review from Golmohammadi an Darvishi. Noise Health. 2019 Jul-Aug; 21(101): 125–141.doi: 10.4103/nah.NAH_4_18

Therefore the question arises whether the search was complete or not. In the enumeration of search terms "Noise-induced hearing loss" or "NIHL" are missing.

The authors should comment or correct these issues and maybe adjust their search terms and include the latest literature

Author Response

Dear reviewer,

We are very grateful to the reviewer for the kind review on our manuscript. Those comments are very helpful for revising and improving our paper.

We have carefully studied the comments and revised the original manuscript, and would like to re-submit it for your consideration. We have addressed the comments raised by the reviewer, and the amendments are highlighted in red in the revised manuscript. Point-by-point responses to your comments are described in attached file.

Reviewer 2 Report

The argument is very interesting, and it’ll be worth publishing after a minor revision. The draft is well conceived; limits are described, but future clinical implications are not (perhaps conclusions could be more incisive, for example, "on the basis of this meta-analysis, we account for very important health surveillance of workers exposed to solvents and noise, and we consider it necessary to carry out... (last The draft is well conceived; limits are described, but future clinical implications are not (perhaps conclusions could be more incisive, for example, "on the basis of this meta-analysis, we account for very important health surveillance of workers exposed to solvents and noise, and we consider it necessary to carry out... (last paragraph)"). Table 3 is not in the correct form to be consulted. Figure 2 It's a little complex and challenging to understand.

Author Response

Dear reviewer,

We are very grateful to the reviewer for the kind review on our manuscript. Those comments are very helpful for revising and improving our paper.

We have carefully studied the comments and revised the original manuscript, and would like to re-submit it for your consideration. We have addressed the comments raised by the reviewer, and the amendments are highlighted in red in the revised manuscript. Point-by-point responses to your comments are described the attachment. 

Reviewer 3 Report

 The updating of literature on this topic is alwalys interesting. The study is well condicted although it could be integrated with more details, tryning to show something more. Tables and figures should be rewritten in a simpler and cleare form.

Author Response

Dear reviewer,

We are very grateful to the reviewer for the kind review on our manuscript. The kind comments are very helpful for revising and improving our paper.

We have carefully studied the comments and revised the original manuscript, and would like to re-submit it for your consideration. We have addressed the comments raised by the reviewer, and the amendments are highlighted in red in the revised manuscript. Point-by-point responses to the reviewer’s comments are described below.

Comment: The updating of literature on this topic is always interesting. The study is well condicted although it could be integrated with more details, trying to show something more. Tables and figures should be rewritten in a simpler and clearer form.

Response: We appreciate the reviewer’s helpful comment. According to the suggestion, the future clinical implications and recommendations to reduce hearing loss have been added in the conclusion section. We suggested that workers simultaneously exposed to noise and solvents should be included in hearing protection programs. Primary prevention, as the most cost-effective measure, such as reducing the level of solvents and noise, and shortening the duration of exposure, should be taken. Furthermore, it is necessary to implement important health surveillance, especially audiometry, for workers exposed to organic solvents and noise. In addition, tables and figures have been rewritten. The results of quality assessment (table 1 and table 2) were attached as supplement information. Table 3 (presented as table 1 and table 2 in revised manuscript) and figure 2 have been simplified.

Once again, thank you very much for your good comments and hope that the explanations will meet your requirements. Looking forward to hearing from you.

With best wishes.
Yours sincerely,

Xiangjing Gao

Reviewer 4 Report

Dear Author/Authors,

Please consider the following recommendations:

Please refine the personal expression from all paper, like “we added” into impersonal expression that ‘it was added”, because it is a scientific paper and it’s understood that the authors have made the work. See the Abstracts and the lines: 101, 131, 251.

Table 1. Please insert lines in the table for easy viewing of the rows;

Table 3: Please reduce the writing in the head of the table so that the words can be whole;

Figure 2. This graph is unclear. Find another approach to present this cumulative information. Hearing loss of more than 250% is not possible. I understood the basic idea of rendering the information presented graphically, but the method of rendering is unrepresentative and unclear.

Supplement the conclusions with recommendations to reduce hearing loss as a result of the exposures specified in the study.

Regards,

Author Response

Dear reviewer,

We are very grateful to the reviewer for the kind review on our manuscript. Those comments are very helpful for revising and improving our paper.

We have carefully studied the comments and revised the original manuscript, and would like to re-submit it for your consideration. We have addressed the comments raised by the reviewer, and the amendments are highlighted in red in the revised manuscript. Point-by-point responses to your comments are described in the attachment. 

Reviewer 5 Report

Well researched paper with appropriate guidelines to review and select the best articles for the meta analysis.    Conclusions seem adequate but with the limitations that it is not clear the duration of the exposure to noise and solvents, what protective mechanisms were in place and utilized by the workers and variable solvents that apparently having different levels of ototoxicity.

Author Response

Dear reviewer,

We are very grateful to the reviewer for the kind review on our manuscript. The kind comments are very helpful for revising and improving our paper.

We have carefully studied the comments and revised the original manuscript, and would like to re-submit it for your consideration. We have addressed the comments raised by the reviewer, and the amendments are highlighted in red in the revised manuscript. Point-by-point responses to your comments are described below.

Comment: Conclusions seem adequate but with the limitations that it is not clear the duration of the exposure to noise and solvents, what protective mechanisms were in place and utilized by the workers and variable solvents that apparently having different levels of ototoxicity.

Response: We appreciate to the helpful comment. The duration of the exposure to noise and solvents in each work shift, protective measurements, and personal protective equipment are influential factors for the risk of hearing loss. Unfortunately, these factors were not available from the included references and their contribution could not be discussed in this study. This limitation has been added in the discussion section (lines 354-357).

Once again, we appreciate for reviewer’s careful consideration, and hope that the explanations will meet your requirements. If the explanations could not meet your requirements, we hope you can give us specific suggestions for further revision. Looking forward to hearing from you.

With best wishes.
Yours sincerely,

Xiangjing Gao

Round 2

Reviewer 1 Report

The authors have substancially  improved the manuscript and expanded their literature research to the current state. They have thus complied with the suggestions.

Reviewer 2 Report

The authors have satisfied the criticism of the initial version and the paper is suitable for publication 

Reviewer 4 Report

Dear Authors, 

Now is Ok, success!

Regards,